# Aristotelian Fragments and Subdiagrams for the Boolean Algebra $\mathbb{B}_5$

**Koen Roelandt**  **and Hans Smessaert** * 

Department of Linguistics, KU Leuven, 3000 Leuven, Belgium; koen.roelandt@kuleuven.be
* Correspondence: hans.smessaert@kuleuven.be

**Abstract:** On a descriptive level, this paper presents a number of logical fragments which require the Boolean algebra $\mathbb{B}_5$, i.e., bitstrings of length five, for their semantic analysis. Two categories from the realm of natural language quantification are considered, namely, proportional quantification with fractions and percentages—as in *two thirds/66 percent of the children are asleep*—and normative quantification—as in *not enough/too many children are asleep*. On a more theoretical level, we study two distinct Aristotelian subdiagrams in $\mathbb{B}_5$, which are the result of moving from $\mathbb{B}_5$ to $\mathbb{B}_4$ either by collapsing bit positions or by deleting bit positions. These two operations are also argued to shed a new light on earlier results from Logical Geometry, in which the collapsing or deletion of bit positions triggers a shift from $\mathbb{B}_4$ to $\mathbb{B}_3$.

**Keywords:** logical geometry; bitstring semantics; Boolean algebras $\mathbb{B}_4/\mathbb{B}_5$; proportional quantification; normative quantification; deontic modality; Aristotelian quantifiers; Aristotelian subdiagrams; rhombic icosahedron; rhombic dodecahedron; hypercube

**MSC:** 03B65; 03G05

## 1. Introduction

In recent work on Logical Geometry, two topics have been studied in great detail, namely, logic sensitivity and Boolean subfamilies of Aristotelian families. In both cases the toolkit of bitstring semantics plays a crucial role in the analysis. A classical Aristotelian square can be captured by bitstrings of length three, whereas a degenerate square requires bitstrings of length four [1]. Similarly, a strong Jacoby–Sesmat–Blanché hexagon can be encoded with bitstrings of length three, whereas its weak counterpart requires length four [1–6]. On the level of octagons, the Aristotelian family of Buridan octagons has Boolean subtypes of bitstring lengths four, five and six [7–11], whereas the Aristotelian family of Keynes–Johnson octagons has Boolean subtypes of bitstring lengths six and seven [12–14]. Finally, logic sensitivity and existential import also distinguish the two octagons studied in the present Special Issue Volume for the interaction between the quantifiers *all* and *most*: the logical system without existential import requires bitstrings of length six, whereas the one with existential import requires bitstings of length five.

The aim of the present paper—in which bitstrings play a central role as well—is twofold. First of all, on a more DESCRIPTIVE level, we want to present a number of logical fragments from the realm of natural language quantification whose semantic analysis requires bitstrings of length five. Secondly, on a more THEORETICAL level, we want to study the notion of Aristotelian subdiagrams in terms of two types of general transformations on bitstrings. The paper is structured as follows. In Section 2, we introduce the basic concepts: bitstrings, the Boolean algebras $\mathbb{B}_3$, $\mathbb{B}_4$ and $\mathbb{B}_5$ and Aristotelian relations within these algebras. In Section 3, we present logical fragments with expressions from two domains of natural language quantification which is given a semantic analysis in terms of $\mathbb{B}_5$, namely, proportional quantification with fractions and percentages—as in *two thirds/66 percent of*

*the children are asleep*—and normative quantification—as in *not enough/too many children are asleep*. In Section 4, we present two distinct Aristotelian subdiagrams in $\mathbb{B}_5$ which are the result of moving from $\mathbb{B}_5$ to $\mathbb{B}_4$ either by collapsing bit positions or by deleting bit positions. In Section 5, we show how these two operations also shed a new light on earlier results from Logical Geometry, in which the collapsing or deletion of bit positions from $\mathbb{B}_4$ to $\mathbb{B}_3$ yields different subdiagrams for $\mathbb{B}_4$. In Section 6, we draw conclusions and point to some prospects for further research.

## 2. Bitstrings, Boolean Algebras and Aristotelian Relations

In Logical Geometry, a BITSTRING is defined as a string of bit values, i.e., a sequence of values 0 or 1, such as 100, 1001 or 11011. In the present paper, we focus on sets of bitstrings $BS_n$ for $3 \leq n \leq 5$, i.e., bitstrings consisting of three, four or five positions:

$$
\begin{aligned}
BS_3 &:= \{0,1\}^3 = \{000, 001, \ldots, 110, 111\} \\
BS_4 &:= \{0,1\}^4 = \{0000, 0001, \ldots 1110, 1111\} \\
BS_5 &:= \{0,1\}^5 = \{00000, 00001, \ldots, 11110, 11111\}
\end{aligned}
$$

A set $BS_n$ contains $2^n$ logically possible combinations: i.e., $|BS_3| = 2^3 = 8$, $|BS_4| = 2^4 = 16$ and $|BS_5| = 2^5 = 32$. Furthermore, the standard BOOLEAN OPERATIONS of complementation ($\neg$), conjunction ($\wedge$) and disjunction ($\vee$) can straightforwardly be defined on bitstrings, as illustrated for some elements of $BS_5$:

| $\neg$ | $\wedge$ | $\vee$ |
|---|---|---|
| | 11**1**00 | 11**0**00 |
| 11000 | 00**1**11 | 00**0**11 |
| = | = | = |
| 00111 | 00**1**00 | 11**0**11 |

Complementation takes a bitstring and returns its complement bitstring with opposite values in all bit positions, whereas conjunction and disjunction take two bitstrings as input and compute the result *bit position by bit position* in the way the propositional connectives do in truth tables. The BOOLEAN ALGEBRA $\mathbb{B}_n$ can then be defined as follows:

$$
\mathbb{B}_n := \langle BS_n, \neg, \wedge, \vee, \top_n, \bot_n \rangle
$$

i.e., as a mathematical structure consisting of six components: (1) the set of bitstrings $BS_n$, the Boolean operations of (2) complementation $\neg$, (3) conjunction $\wedge$ and (4) disjunction $\vee$, and the two special elements (5) top $\top_n$ and (6) bottom $\bot_n$. The latter two are the bitstrings exclusively consisting of values 1 and those exclusively consisting values 0, respectively:

$$
\begin{aligned}
\top_3 &:= 111 & \top_4 &:= 1111 & \top_5 &:= 11111 \\
\bot_3 &:= 000 & \bot_4 &:= 0000 & \bot_5 &:= 00000
\end{aligned}
$$

In the framework of Logical Geometry [1,15], a central object of investigation is the so-called 'Aristotelian square' or 'square of opposition', visualising ARISTOTELIAN RELATIONS, i.e., logical relations of opposition and implication. Informally, two propositions $\alpha$ and $\beta$ are said to be

| | | | |
|---|---|---|---|
| contradictory | $CD(\alpha,\beta)$ | iff | $\alpha$ and $\beta$ cannot be true together and $\alpha$ and $\beta$ cannot be false together, |
| contrary | $CR(\alpha,\beta)$ | iff | $\alpha$ and $\beta$ cannot be true together but $\alpha$ and $\beta$ can be false together, |
| subcontrary | $SCR(\alpha,\beta)$ | iff | $\alpha$ and $\beta$ can be true together but $\alpha$ and $\beta$ cannot be false together, |
| in subalternation | $SA(\alpha,\beta)$ | iff | $\alpha$ entails $\beta$ but $\beta$ does not entail $\alpha$. |

These Aristotelian relations—as defined between formulas—can be straightforwardly 'reformulated' in terms of Boolean operations on bitstrings. The bitstrings $b_1$ and $b_2 \in BS_n$ are said to be

$$
\begin{array}{llll}
\textit{n-contradictory} & \text{iff} & b_1 \wedge b_2 = \bot_n & \text{and} & b_1 \vee b_2 = \top_n, \\
\textit{n-contrary} & \text{iff} & b_1 \wedge b_2 = \bot_n & \text{and} & b_1 \vee b_2 \neq \top_n, \\
\textit{n-subcontrary} & \text{iff} & b_1 \wedge b_2 \neq \bot_n & \text{and} & b_1 \vee b_2 = \top_n, \\
\textit{in n-subalternation} & \text{iff} & b_1 \wedge b_2 = b_1 & \text{and} & b_1 \vee b_2 \neq b_1.
\end{array}
$$

As demonstrated below, the two bitstrings 11000, 00111 $\in BS_5$ are 5-contradictory since their conjunction equals $\bot_5$—there is no 'overlap' in any bit position—and their disjunction equals $\top_5$—there is no 'gap' in any bit position either. The bitstrings 11000, 00011 $\in BS_5$, by contrast, are 5-contrary since their conjunction again equals $\bot_5$—there is no 'overlap'—but their disjunction does not equal $\top_5$—there is indeed a 'gap' in the third bit position:

| contradiction | | contrariety | |
|:---:|:---:|:---:|:---:|
| $\wedge$ | $\vee$ | $\wedge$ | $\vee$ |
| 11000 | 11000 | 11000 | 11000 |
| 00111 | 00111 | 00011 | 00011 |
| 00000 | 11111 | 00000 | 11011 |

## 3. Fragments for the Boolean Algebra $\mathbb{B}_5$

In Logical Geometry, bitstrings—in particular (but not only) those from the Boolean algebras $\mathbb{B}_3$, $\mathbb{B}_4$ and $\mathbb{B}_5$ introduced in Section 2—have been used as compact representations of the denotations of logical formulas and various fragments of natural language expressions. Thus, the $\mathbb{B}_3$ structure underlies, among others, the classical Aristotelian square and its extension to the (strong) Jacoby–Sesmat–Blanché hexagon [1,6]. The $\mathbb{B}_4$ structure underlies, among others, the propositional connectives and the modal logic S5 [16,17], and the $\mathbb{B}_5$ structure underlies, among others, the negative *un*-prefixation with scalar adjectives, as in *not (un)wise* [18]. In the present section, we will present a number of logical fragments with expressions from two domains of natural language quantification that are also analysable in terms of $\mathbb{B}_5$, namely, that of proportional quantification (Section 3.1) and that of normative quantification (Section 3.2).

### 3.1. Proportional Quantification in $\mathbb{B}_5$

Two standard ways to express the notion of proportional quantification in natural language are by means of fractions (such as *more than one third/at least three quarters*) or percentages (such as *exactly seventy five percent*). On its standard reading in Generalized Quantifier Theory [19–21], the sentence *at least $\frac{3}{4}$ of the A's are B* is true iff the number of A's that are B is greater than or equal to $\frac{3}{4}$ of the number of A's. Starting from this basic formula *at least $\frac{3}{4}$(A,B)*, we can then negate either the complete formula, or the predicate B, or both. This yields a first fragment for the system of PROPORTIONAL QUANTIFICATION (PQ), i.e., $\mathcal{F}_1$, which is listed here, together with the formulas' denotations in the standard set-theoretical notation format of Generalized Quantifier Theory (GQT):

$$
\mathcal{F}_1 \quad := \quad \{
\begin{array}{ll}
\text{at least } \tfrac{3}{4}(A,B), & |A \cap B| \geq \tfrac{3}{4}|A| \\
\text{less than } \tfrac{3}{4}(A,B), & |A \cap B| < \tfrac{3}{4}|A| \\
\text{at least } \tfrac{3}{4}(A,\neg B), & |A \setminus B| \geq \tfrac{3}{4}|A| \\
\text{less than } \tfrac{3}{4}(A,\neg B) \quad \} & |A \setminus B| < \tfrac{3}{4}|A|
\end{array}
$$

Now one crucial property of proportional quantifiers is the relation of COMPLEMENTARITY between a given fraction $\frac{3}{4}$ or percentage 75% and its complement fraction $\frac{1}{4}$ or percentage 25%, i.e., what you need to add to the original fraction or percentage to obtain $\frac{4}{4} = 1$ or 100%, respectively. We will henceforth informally refer to the original fraction $\frac{3}{4}$ as the 'large' fraction, and to its complement $\frac{1}{4}$ as the 'small' fraction. From the logical equivalence between the proposition *at least $\frac{3}{4}$ of the students passed the test* and *at most $\frac{1}{4}$ of the students failed the test*, we can now infer the particular interaction between the quantifiers, the complementarity of the fractions, and the predicate negation. In particular, *at least*

$\frac{3}{4}(A,B) \equiv$ *at most* $\frac{1}{4}(A, \neg B)$. This allows us to (1) reformulate the fragment $\mathcal{F}_1$ above as the first four formulas of the fragment $\mathcal{F}_{1'}$ below, and to (2) expand the original fragment by adding two formulas which crucially involve Boolean combinations of the large and small fractions:

$$
\begin{array}{lll}
\mathcal{F}_{1'} \quad := \quad \{ & \text{at least } \frac{3}{4}(A,B), & |A \cap B| \geq \frac{3}{4}|A| \\
& \text{less than } \frac{3}{4}(A,B), & |A \cap B| < \frac{3}{4}|A| \\
& \text{at most } \frac{1}{4}(A,B), & |A \cap B| \leq \frac{1}{4}|A| \\
& \text{more than } \frac{1}{4}(A,B), & |A \cap B| > \frac{1}{4}|A| \\
& \text{between } \frac{1}{4} \text{ and } \frac{3}{4}(A,B), & \frac{3}{4}|A| \geq |A \cap B| \geq \frac{1}{4}|A| \\
& \text{more than } \frac{3}{4} \text{ or less than } \frac{1}{4}(A,B) \quad \} & |A \cap B| > \frac{3}{4}|A| \vee |A \cap B| < \frac{1}{4}|A|
\end{array}
$$

Let us now turn to the partition $\Pi_{\mathsf{PQ}}$ induced by this fragment $\mathcal{F}_{1'}$, consisting of five ANCHOR FORMULAS $\alpha_n$ (the technical procedure to generate this type of partition on the basis of a particular logical fragment is described in full detail in [1]):

$$
\begin{array}{lll}
\Pi_{\mathsf{PQ}}(\mathcal{F}_{1'}) = \{ & \alpha_1: \text{more than } \frac{3}{4}(A,B), & |A \cap B| > \frac{3}{4}|A| \\
& \alpha_2: \text{exactly } \frac{3}{4}(A,B), & |A \cap B| = \frac{3}{4}|A| \\
& \alpha_3: \text{less than } \frac{3}{4} \text{ but more than } \frac{1}{4}(A,B), & \frac{3}{4}|A| > |A \cap B| > \frac{1}{4}|A| \\
& \alpha_4: \text{exactly } \frac{1}{4}(A,B), & |A \cap B| = \frac{1}{4}|A| \\
& \alpha_5: \text{less than } \frac{1}{4}(A,B) \quad \} & |A \cap B| < \frac{1}{4}|A|
\end{array}
$$

In a second step, the bitstring semantics is defined, not just for the fragment $\mathcal{F}_{1'}$ itself, but rather for its entire BOOLEAN CLOSURE in PQ, denoted $\mathbb{B}_{\mathsf{PQ}}(\mathcal{F}_{1'})$ and defined as the smallest set $C \subseteq \mathcal{L}_{\mathsf{PQ}}$ such that (i) $\mathcal{F}_{1'} \subseteq C$ and (ii) $C$ is closed under the Boolean operations (up to logical equivalence), i.e., for all $\varphi, \psi \in C$, there exist $\alpha, \beta \in C$ such that $\alpha \equiv_{\mathsf{PQ}} \varphi \wedge \psi$ and $\beta \equiv_{\mathsf{PQ}} \neg \varphi$. The bitstring semantics $\beta_{\mathsf{PQ}}$ maps every formula $\varphi \in \mathbb{B}_{\mathsf{PQ}}(\mathcal{F}_{1'})$ onto its bitstring representation $\beta_{\mathsf{PQ}}(\varphi)$, which is a sequence of bits that will have the value 1 in its $i$th bit position iff $\models_{\mathsf{PQ}} \alpha_i \rightarrow \varphi$. Given that $|\Pi_{\mathsf{PQ}}(\mathcal{F}_{1'})| = 5$, the BITSTRING SEMANTICS $\beta_{\mathsf{PQ}}$ for $\mathbb{B}_{\mathsf{PQ}}(\mathcal{F}_{1'})$ is defined in terms of the Boolean algebra $\mathbb{B}_5$, i.e., bitstrings of length five. In particular, the resulting bitstrings for the formulas of $\mathcal{F}_{1'}$ are

$$
\begin{array}{llll}
\beta_{\mathsf{PQ}}(\text{at least } \frac{3}{4}(A,B)) & = & 11000 & |A \cap B| \geq \frac{3}{4}|A| \\
\beta_{\mathsf{PQ}}(\text{less than } \frac{3}{4}(A,B)) & = & 00111 & |A \cap B| < \frac{3}{4}|A| \\
\beta_{\mathsf{PQ}}(\text{at most } \frac{1}{4}(A,B)) & = & 00011 & |A \cap B| \leq \frac{1}{4}|A| \\
\beta_{\mathsf{PQ}}(\text{more than } \frac{1}{4}(A,B)) & = & 11100 & |A \cap B| > \frac{1}{4}|A| \\
\beta_{\mathsf{PQ}}(\text{between } \frac{1}{4} \text{ and } \frac{3}{4}(A,B)) & = & 01110 & \frac{3}{4}|A| \geq |A \cap B| \geq \frac{1}{4}|A| \\
\beta_{\mathsf{PQ}}(\text{more than } \frac{3}{4} \text{ or less than } \frac{1}{4}(A,B)) & = & 10001 & |A \cap B| > \frac{3}{4}|A| \vee |A \cap B| < \frac{1}{4}|A|
\end{array}
$$

Since the bitstring 11000 assigned to *at least* $\frac{3}{4}$ contains two values 1, it is referred to as a LEVEL-2 bitstring. It corresponds to the disjunction of the first two anchor formulas in the partition $\Pi_{\mathsf{PQ}}(\mathcal{F}_{1'})$, i.e., $\alpha_1 \vee \alpha_2$, and thus reflects the disjunctive semantics of *at least* $\frac{3}{4}$ as *exactly or more than* $\frac{3}{4}$. Completely analogously, the 00011 bitstring for *at most* $\frac{1}{4}$ expresses the disjunction $\alpha_4 \vee \alpha_5$, i.e., *exactly or less than* $\frac{1}{4}$. Similarly, the level-3 bitstring 01110 for *between* $\frac{1}{4}$ *and* $\frac{3}{4}$ captures the disjunction of the middle three anchor formulas $\alpha_2 \vee \alpha_3 \vee \alpha_4$, namely, (*exactly* $\frac{3}{4}$) or (*less than* $\frac{3}{4}$ *but more than* $\frac{1}{4}$) or (*exactly* $\frac{1}{4}$).

It can now easily be demonstrated that the first four formulas in $\mathcal{F}_{1'}$ constitute a classical Aristotelian square: *at least* $\frac{3}{4}$ (11000) and *less than* $\frac{3}{4}$ (00111) are contradictory, and so are *at most* $\frac{1}{4}$ (00011) and *more than* $\frac{1}{4}$ (11100). In addition, *at least* $\frac{3}{4}$ (11000) and *at most* $\frac{1}{4}$ (00011) are contrary, whereas *less than* $\frac{3}{4}$ (00111) and *more than* $\frac{1}{4}$ (11100) are subcontrary. Notice that these same four formulas/bitstrings can also be shown to yield a so-called DUALITY SQUARE—in terms of logical relations of external, internal and dual

negation—but, as was argued in full detail in [22], it is crucial to keep in mind the logical independence of Aristotelian and duality notions.

The scalar structure underlying the pentapartition of $\Pi_{\mathsf{PQ}}(\mathcal{F}_{1'})$ is visualised in Figure 1a, where the two precise indications of the large fraction for $\alpha_2$ and its complementary small fraction for $\alpha_4$ are represented as two points on the scale, whereas the other three parts of logical space—i.e., $\alpha_1$, $\alpha_3$ and $\alpha_5$—are represented as intervals on the scale.

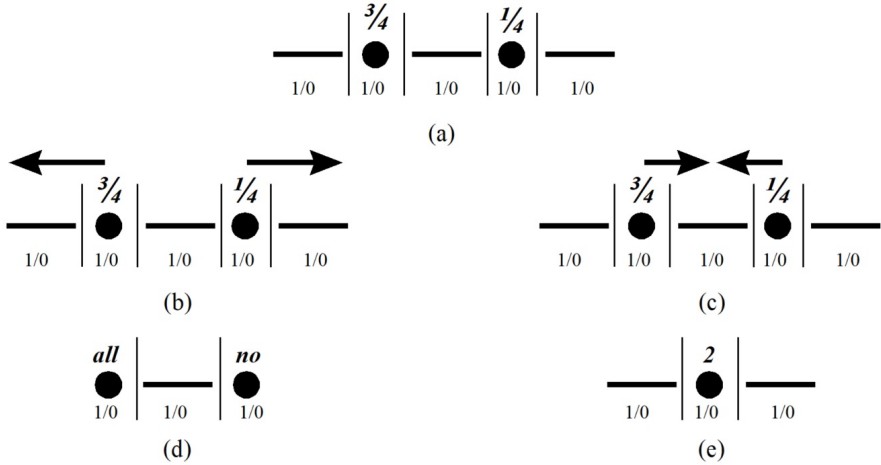

**Figure 1.** (**a**) Scalar structure for proportional quantifiers in $\mathbb{B}_5$ (**b**) moving complementary fractions outwards (**c**) moving complementary fractions inwards (**d**) collapsing into tripartition for standard quantifiers in $\mathbb{B}_3$ (**e**) collapsing into tripartition for numerical quantifiers in $\mathbb{B}_3$.

Starting from this basic constellation in Figure 1a, there are two general strategies for changing the location of the two 'points' on the scale, both of which of course have to respect the property of complementarity, i.e., the large fraction and the small fraction will always have to be moved in *opposite* directions. On the first strategy—visually represented in Figure 1b—the complementary fractions are moving OUTWARD: the large fraction moves to the left in the direction of $\frac{4}{4}$, whereas the small fraction moves to the right in the direction of $\frac{0}{4}$, resulting in the new fragment $\mathcal{F}_2$:

$$\mathcal{F}_2 \quad := \quad \{$$

| | |
|---|---|
| at least $\frac{4}{4}$(A,B), | $\lvert A \cap B \rvert \geq \frac{4}{4} \lvert A \rvert$ |
| less than $\frac{4}{4}$(A,B), | $\lvert A \cap B \rvert < \frac{4}{4} \lvert A \rvert$ |
| at most $\frac{0}{4}$(A,B), | $\lvert A \cap B \rvert \leq \frac{0}{4} \lvert A \rvert$ |
| more than $\frac{0}{4}$(A,B), | $\lvert A \cap B \rvert > \frac{0}{4} \lvert A \rvert$ |
| between $\frac{0}{4}$ and $\frac{4}{4}$(A,B), | $\frac{4}{4} \lvert A \rvert \geq \lvert A \cap B \rvert \geq \frac{0}{4} \lvert A \rvert$ |
| more than $\frac{4}{4}$ or less than $\frac{0}{4}$(A,B)  } | $\lvert A \cap B \rvert > \frac{3}{4} \lvert A \rvert \vee \lvert A \cap B \rvert < \frac{1}{4} \lvert A \rvert$ |

In a first step, we can now generate the partition $\Pi_{\mathsf{PQ}}(\mathcal{F}_2)$ induced by this new fragment, as a simple modification of the original pentapartition:

$$\Pi_{\mathsf{PQ}}(\mathcal{F}_2) = \{$$

| | | |
|---|---|---|
| $\alpha_1^*$: more than $\frac{4}{4}$(A,B), | $\lvert A \cap B \rvert > \frac{4}{4} \lvert A \rvert$ |
| $\alpha_2^*$: exactly $\frac{4}{4}$(A,B), | $\lvert A \cap B \rvert = \frac{4}{4} \lvert A \rvert$ |
| $\alpha_3^*$: less than $\frac{4}{4}$ but more than $\frac{0}{4}$(A,B), | $\frac{0}{4} \lvert A \rvert < \lvert A \cap B \rvert < \frac{4}{4} \lvert A \rvert$ |
| $\alpha_4^*$: exactly $\frac{0}{4}$(A,B), | $\lvert A \cap B \rvert = \frac{0}{4} \lvert A \rvert$ |
| $\alpha_5^*$: less than $\frac{0}{4}$(A,B)  } | $\lvert A \cap B \rvert < \frac{0}{4} \lvert A \rvert$ |

In theory, this partition would again give rise to a new bitstring semantics in terms of $\mathbb{B}_5$, i.e., bitstrings of length five. However, no value can be assigned to $\alpha_1^*$ and $\alpha_5^*$ since both *more than* $\frac{4}{4}$(A,B) and *less than* $\frac{0}{4}$(A,B) are contradictory formulas:

$$\beta_{\mathsf{PQ}}(\text{at least } \tfrac{4}{4}(A,B)) \qquad\qquad = \quad \text{-100-} \quad |A \cap B| \geq \tfrac{4}{4}|A|$$

$$\beta_{\mathsf{PQ}}(\text{less than } \tfrac{4}{4}(A,B)) \qquad\qquad = \quad \text{-011-} \quad |A \cap B| < \tfrac{4}{4}|A|$$

$$\beta_{\mathsf{PQ}}(\text{at most } \tfrac{0}{4}(A,B)) \qquad\qquad = \quad \text{-001-} \quad |A \cap B| \leq \tfrac{0}{4}|A|$$

$$\beta_{\mathsf{PQ}}(\text{more than } \tfrac{0}{4}(A,B)) \qquad\qquad = \quad \text{-110-} \quad |A \cap B| > \tfrac{0}{4}|A|$$

$$\beta_{\mathsf{PQ}}(\text{between } \tfrac{0}{4} \text{ and } \tfrac{4}{4}(A,B)) \qquad = \quad \text{-111-} \quad \tfrac{4}{4}|A| \geq |A \cap B| \geq \tfrac{0}{4}|A|$$

$$\beta_{\mathsf{PQ}}(\text{more than } \tfrac{4}{4} \text{ or less than } \tfrac{0}{4}(A,B)) = \quad \text{-000-} \quad |A \cap B| > \tfrac{3}{4}|A| \vee |A \cap B| < \tfrac{1}{4}|A|$$

Notice that the last two formulas—with their Boolean combinations of a large and a small fraction—have become non-contingent: *between $\tfrac{0}{4}$ and $\tfrac{4}{4}(A,B)$* is always true, whereas *more than $\tfrac{4}{4}$ or less than $\tfrac{0}{4}(A,B)$* is always false. In this respect, the fragment $\mathcal{F}_2$ violates the second of the two standard assumptions for fragments in Logical Geometry, namely, (1) that they be closed under negation (which is still the case) and (2) that they only contain contingent formulas. As illustrated in the transition from Figure 1b to Figure 1d above, the collapsing of the bit positions for $\alpha_1$ and $\alpha_2$ at the extreme left end of the scalar structure, and of those for $\alpha_4$ and $\alpha_5$ at the extreme right end, yield a tripartition. This tripartition can now easily be shown to underly the quantifiers of the logical system for SYLLOGISTICS (SYL), i.e., the standard quantifiers of predicate logic with existential import [1]. After the elimination of the two non-contingent formulas, the fragment $\mathcal{F}_2$ can thus be reformulated as the four formula fragment $\mathcal{F}_{2'}$:

$$\mathcal{F}_{2'} \quad := \quad \{ \qquad \text{all}(A,B), \qquad\qquad |A \cap B| = |A|$$
$$\text{not all}(A,B), \qquad\qquad |A \cap B| < |A|$$
$$\text{no}(A,B), \qquad\qquad |A \cap B| = 0$$
$$\text{some}(A,B) \quad \} \qquad |A \cap B| > 0$$

This fragment induces the partition $\Pi_{\mathsf{SYL}}(\mathcal{F}_{2'})$—visually represented as the scalar structure in Figure 1d—which underlies the bitstring semantics $\beta_{\mathsf{SYL}}$ in terms of $\mathbb{B}_3$, i.e., bitstrings of length three:

$$\Pi_{\mathsf{SYL}}(\mathcal{F}_{2'}) = \{ \quad \alpha_2': \text{all}(A,B), \qquad\qquad |A \cap B| = |A|$$
$$\alpha_3': \text{some but not all}(A,B), \qquad 0 < |A \cap B| < |A|$$
$$\alpha_4': \text{no}(A,B) \qquad\qquad \} \quad |A \cap B| = 0$$

$$\beta_{\mathsf{SYL}}(\text{all}(A,B)) \qquad = \quad 100 \quad |A \cap B| = |A|$$
$$\beta_{\mathsf{SYL}}(\text{not all}(A,B)) \quad = \quad 011 \quad |A \cap B| < |A|$$
$$\beta_{\mathsf{SYL}}(\text{no}(A,B)) \qquad = \quad 001 \quad |A \cap B| = 0$$
$$\beta_{\mathsf{SYL}}(\text{some}(A,B)) \quad = \quad 110 \quad |A \cap B| > 0$$

Starting from the basic constellation in Figure 1a, the second strategy for changing the location of the two 'points' on the scale—visually represented in Figure 1c—moves the complementary fractions INWARD. This again respects the property of complementarity, according to which the large and small fractions have to be moved in opposite directions: the large fraction moves to the right and the small one to the left, in order for them to coincide in the 'exactly half' fraction in the center of the structure. This results in the new fragment $\mathcal{F}_3$:

$$\mathcal{F}_3 \quad := \quad \{ \qquad\qquad\qquad \text{at least } \tfrac{2}{4}(A,B), \qquad |A \cap B| \geq \tfrac{2}{4}|A|$$
$$\text{less than } \tfrac{2}{4}(A,B), \qquad |A \cap B| < \tfrac{2}{4}|A|$$
$$\text{at most } \tfrac{2}{4}(A,B), \qquad |A \cap B| \leq \tfrac{2}{4}|A|$$
$$\text{more than } \tfrac{2}{4}(A,B), \qquad |A \cap B| > \tfrac{2}{4}|A|$$
$$\text{between } \tfrac{2}{4} \text{ and } \tfrac{2}{4}(A,B), \qquad \tfrac{2}{4}|A| \geq |A \cap B| \geq \tfrac{2}{4}|A|$$
$$\text{more than } \tfrac{2}{4} \text{ or less than } \tfrac{2}{4}(A,B) \quad \} \qquad |A \cap B| > \tfrac{2}{4}|A| \vee |A \cap B| < \tfrac{2}{4}|A|$$

The bit positions for $\alpha_2$, $\alpha_3$ and $\alpha_4$ in the original pentapartition $\Pi_{PQ}(\mathcal{F}_{1'})$ above collapse into a single bit position in the centre of the scalar structure. The resulting tripartition—$\Pi_{PQ}(\mathcal{F}_3)$—underlies the bitstring semantics $\beta'_{PQ}$ in terms of $\mathbb{B}_3$, i.e., bitstrings of length three:

$$\Pi_{PQ}(\mathcal{F}_3) = \{ \quad \begin{array}{lll} \alpha''_1\text{: more than } \frac{2}{4}\text{(A,B),} & & |A \cap B| > \frac{2}{4}|A| \\ \alpha''_3\text{: exactly } \frac{2}{4}\text{(A,B),} & & |A \cap B| = \frac{2}{4}|A| \\ \alpha''_5\text{: less than } \frac{2}{4}\text{(A,B)} & \} & |A \cap B| < \frac{2}{4}|A| \end{array}$$

$$
\begin{array}{llll}
\beta'_{PQ}(\text{at least } \tfrac{2}{4}(A,B)) & = & 110 & |A \cap B| \geq \tfrac{2}{4}|A| \\
\beta'_{PQ}(\text{less than } \tfrac{2}{4}(A,B)) & = & 001 & |A \cap B| < \tfrac{2}{4}|A| \\
\beta'_{PQ}(\text{at most } \tfrac{2}{4}(A,B)) & = & 011 & |A \cap B| \leq \tfrac{2}{4}|A| \\
\beta'_{PQ}(\text{more than } \tfrac{2}{4}(A,B)) & = & 100 & |A \cap B| > \tfrac{2}{4}|A| \\
\beta'_{PQ}(\text{between } \tfrac{2}{4} \text{ and } \tfrac{2}{4}(A,B)) & = & 010 & \tfrac{2}{4}|A| \geq |A \cap B| \geq \tfrac{2}{4}|A| \\
\beta'_{PQ}(\text{more than } \tfrac{2}{4} \text{ or less than } \tfrac{2}{4}(A,B)) & = & 101 & |A \cap B| > \tfrac{2}{4}|A| \vee |A \cap B| < \tfrac{2}{4}|A|
\end{array}
$$

Observe, first of all, that—in contrast to the last two formulas in the fragment $\mathcal{F}_2$ above—the last two formulas in $\mathcal{F}_3$ have *not* become non-contingent: *between $\frac{2}{4}$ and $\frac{2}{4}$(A,B) $\equiv$ exactly half(A,B)* and *more than $\frac{2}{4}$ or less than $\frac{2}{4}$(A,B) $\equiv$ not exactly half(A,B)*. Secondly, this analysis of the six formulas in $\mathcal{F}_3$ is isomorphic to that for the six formulas that standardly show up in the realm of numerical (but *non*-proportional) quantification, namely, *more/less than 2(A,B)*, *at least/most 2(A,B)* or *(not) exactly 2(A,B)*, the scalar structure of which is visualised in Figure 1e. Thirdly, the tripartition $\Pi_{PQ}(\mathcal{F}_3)$ also plays a crucial role in the construction of the two octagons studied in the present Special Issue Volume for the interaction between the quantifiers *all* and *most* [23].

### 3.2. Normative Quantification in $\mathbb{B}_5$

We now turn to a second domain of natural language quantification, namely, that of normative expressions such as *(not) enough* and *too many/few*. Both conceptually and technically, the analysis of these normative quantifiers turns out to be very closely related to that of the proportional quantifiers in Section 3.1 in terms of the pentapartition of $\mathbb{B}_5$. Furthermore, these quantifiers allow us to establish the precise connection that this Special Issue is dedicated to, namely, that between Modal Logic and Logical Geometry. In particular, the semantics of the quantifiers *(not) enough* and *(not) too many* crucially involves the deontic modal notions of '(minimal) amount required' and '(maximal) amount allowed'. Suppose we want to go on a sailing trip and we need at least four people to sail the boat, but the boat can carry at most eight people. In this context, the proposition *Not enough people showed up for the sailing trip* is true if the number of people that actually showed up is smaller than the minimal number required for the sailing. Similarly, the proposition *Too many people showed up for the sailing trip* is true if the number of people that actually showed up is greater than the maximal number allowed for the sailing.

In order to capture this 'deontic quantification', we first of all expand the standard GQT toolkit with two deontic operators taking scope over one-place predicates. Thus, in addition to $|A \cap B|$ for the number of A's that are actually B, we define $|A \cap \blacksquare(B)|$ as the number of A's that are required to be B and $|A \cap \blacklozenge(B)|$ as the number of A's that are allowed to be B. Secondly, in addition to $|A|$ for the actual number of A's, we add the operators $min|A|$ and $max|A|$ for the minimal and maximal number of A's, respectively. Thirdly, in the realm of normative quantification, the numerical operators *min/max* and the deontic operators $\blacksquare$/$\blacklozenge$ interact in a very specific way, in the sense that the notion of the lower boundary combines *min* with the $\blacksquare$ of obligation, whereas that of the upper boundary combines *max* with the $\blacklozenge$ of permission. This allows us to define a new fragment for the system of NORMATIVE QUANTIFICATION (NQ), i.e., $\mathcal{F}_4$, which is listed here, together with the formulas' denotations in the 'extended' set-theoretical notation format of GQT:

$$\mathcal{F}_4 \quad := \quad \{ \qquad\qquad \text{too many(A,B)}, \qquad |A \cap B| > max|A \cap \blacklozenge(B)|$$

$$\text{not too many(A,B)}, \qquad |A \cap B| \leq max|A \cap \blacklozenge(B)|$$

$$\text{not enough(A,B)}, \qquad |A \cap B| < min|A \cap \blacksquare(B)|$$

$$\text{enough(A,B)}, \qquad |A \cap B| \geq min|A \cap \blacksquare(B)|$$

$$\text{enough but not too many(A,B)}, \qquad max|A \cap \blacklozenge(B)| \geq |A \cap B|$$

$$\geq min|A \cap \blacksquare(B)|$$

$$\text{too many or too few(A,B)} \quad \} \qquad |A \cap B| > max|A \cap \blacklozenge(B)|$$

$$\vee |A \cap B| < min|A \cap \blacksquare(B)|$$

Conceptually speaking, there is a clear similarity between the two 'points' $\alpha_2$ and $\alpha_4$ on the proportional scale—for the large and small proportion respectively—on the one hand, and the normative contrast between an upper boundary $\alpha_2'$ for '(exactly) maximally admissible' and a lower boundary $\alpha_4'$ for '(exactly) minimally required'. Hence, the normative pentapartition $\Pi_{\mathsf{NQ}}$ for the fragment $\mathcal{F}_4$ appears as follows:

$$\Pi_{\mathsf{NQ}}(\mathcal{F}_4) = \{ \quad \alpha_1': \text{too many(A,B)}, \qquad |A \cap B| > max|A \cap \blacklozenge(B)|$$

$$\alpha_2': \text{just not too many(A,B)}, \qquad |A \cap B| = max|A \cap \blacklozenge(B)|$$

$$\alpha_3': \text{not just not too many}$$

$$\text{but not just enough(A,B)}, \qquad max|A \cap \blacklozenge(B)| > |A \cap B|$$

$$> min|A \cap \blacksquare(B)|$$

$$\alpha_4': \text{just enough(A,B)}, \qquad |A \cap B| = min|A \cap \blacksquare(B)|$$

$$\alpha_5': \text{not enough(A,B)} \qquad \} \quad |A \cap B| < min|A \cap \blacksquare(B)|$$

Given this pentapartition, the bitstring semantics $\beta_{\mathsf{NQ}}$ for the Boolean closure of the fragment—$\mathbb{B}_{\mathsf{NQ}}(\mathcal{F}_4)$—is defined in terms of $\mathbb{B}_5$, i.e., bitstrings of length five. For the actual formulas of $\mathcal{F}_4$, the resulting bitstrings are the following:

$$\beta_{\mathsf{NQ}}(\text{too many(A,B)}) \qquad = \quad 10000 \quad |A \cap B| > max|A \cap \blacklozenge(B)|$$

$$\beta_{\mathsf{NQ}}(\text{not too many(A,B)}) \qquad = \quad 01111 \quad |A \cap B| \leq max|A \cap \blacklozenge(B)|$$

$$\beta_{\mathsf{NQ}}(\text{not enough(A,B)}) \qquad = \quad 00001 \quad |A \cap B| < min|A \cap \blacksquare(B)|$$

$$\beta_{\mathsf{NQ}}(\text{enough(A,B)}) \qquad = \quad 11110 \quad |A \cap B| \geq min|A \cap \blacksquare(B)|$$

$$\beta_{\mathsf{NQ}}(\text{enough but not too many(A,B)}) \quad = \quad 01110 \quad max|A \cap \blacklozenge(B)| \geq |A \cap B|$$

$$\geq min|A \cap \blacksquare(B)|$$

$$\beta_{\mathsf{NQ}}(\text{too many or too few(A,B)}) \qquad = \quad 10001 \quad |A \cap B| > max|A \cap \blacklozenge(B)|$$

$$\vee |A \cap B| < min|A \cap \blacksquare(B)|$$

In Section 3.1, we describe two strategies which yielded a reduction or collapse from the pentapartition of $\mathbb{B}_5$ to the tripartition of $\mathbb{B}_3$ in the realm of proportional quantification. On the one hand, the two proportional points on the scale could be moved 'outward'—away from one another—so as to ultimately reach the trivial proportions $\frac{n}{n}$ and $\frac{0}{n}$ at the highest and lowest extreme ends of the scalar structure, as visualised in Figure 1b or Figure 1d. On the other hand, the two complementary proportional points could be moved 'inward'—towards one another—so as to ultimately coincide at the 'halfway' proportion $\frac{n}{2n}$ in the center of the scale, as visualised in Figure 1c or Figure 1e.

Although both strategies are also available in principle for the $\mathbb{B}_5$ structure underlying the realm of normative quantification, there is one crucial difference between the two types of quantification. With the proportional quantifiers, the large and small proportions are by necessity connected to one another through their relationship of complementarity. With the normative quantifiers, by contrast, the upper and lower boundary are independent in principle, or at least do not stand in the very strong relationship of complementarity.

As for the strategy of moving the normative boundary points outward, it is thus perfectly possible to move one of the boundaries outward without moving the other. As a consequence, the possible collapse of the $\alpha_1'$ interval beyond the $\alpha_2'$ upper boundary is logically independent of the collapse of the $\alpha_5'$ interval beyond the $\alpha_4'$ lower boundary.

One could argue, for instance, that with a proposition such as *Too many people care for the planet* the $\alpha'_1$ interval should be eliminated, since everybody should care for the planet, i.e., it should not be possible to surpass any maximal number of people allowed to care for the planet. Unlike with the proportional quantifiers, however, this elimination of the $\alpha'_1$ interval does not automatically trigger that of the $\alpha'_5$ interval beyond the $\alpha'_4$ lower boundary, since—sadly enough, indeed—*Not enough people care for the planet*. Conversely, with the proposition *Not enough people ignore the global water shortage*, the desired elimination of the $\alpha'_5$ interval—i.e., the impossibility to go below any minimal number of people required to ignore—does not automatically trigger that of the $\alpha'_1$ interval beyond the $\alpha'_2$ upper boundary, since—sadly enough again—*Too many people do indeed ignore the global water shortage*. What these two examples reveal is that in the case of normative quantification, the reduction is *not* automatically from $\mathbb{B}_5$ to $\mathbb{B}_3$, but can also be to $\mathbb{B}_4$, when only $\alpha'_1$ or only $\alpha'_5$ is eliminated. Precisely such reductions from $\mathbb{B}_5$ to $\mathbb{B}_4$ will play a crucial role in Section 4—albeit on a somewhat more abstract level.

As for the second reduction strategy—namely, that of moving the normative boundary points inwards in order for the upper and lower boundaries to coincide in the middle of the scalar structure—the situation turns out to be basically identical to that with the proportional quantifiers in Section 3.1. With coinciding upper and lower boundaries, the three central components of the pentapartition $\Pi_{NQ}(\mathcal{F}_4)$—i.e., the two points $\alpha'_2$ and $\alpha'_4$ and the $\alpha'_3$ interval in between—collapse into one central component, and the reduction is again from $\mathbb{B}_5$ to $\mathbb{B}_3$. This situation arises when one and the same exact number *simultaneously* counts as *not too few* (i.e., 'required as minimum') and *not too many* (i.e., 'allowed as maximum'). A standard case in point would be the exact number of players that is both allowed and required to be on the field with each team in a sports competition.

## 4. Aristotelian Subdiagrams for $\mathbb{B}_5$

In the present section, we move from the more descriptive level of characterising logical fragments and their bitstring semantics in $\mathbb{B}_5$ (for various categories of natural language quantifier expressions) to a more theoretical analysis. Two general operations on bitstrings from $\mathbb{B}_5$ will be compared, namely, that of COLLAPSING bit positions (Section 4.1) and that of DELETING bit positions (Section 4.2) [1]. In both cases, the resulting move from $\mathbb{B}_5$ to $\mathbb{B}_4$ is characterised in terms of SUBDIAGRAMS of $\mathbb{B}_5$.

This notion of subdiagram obviously relates to the strong concern in Logical Geometry with the visualisation of the logical relations in Aristotelian diagrams. So, before turning to the two operations on $\mathbb{B}_5$ bitstrings, let us first briefly consider this visualisation aspect. An important contrast made in Logical Geometry is that between Hasse diagrams and Aristotelian or logical diagrams [17]. With Hasse diagrams—which constitute a standard visualisation of Boolean algebras [24]—the focus is primarily on implication relations, and the top and bottom elements occupy prominent positions at the extreme (top and bottom) ends of the diagram. With logical diagrams, by contrast, the focus is primarily on opposition relations, and the top and bottom elements are considered as 'trivial' elements (tautology and contradiction). Strictly speaking, these top and bottom elements coincide in the center of the diagram, but they are hardly ever visually represented. As a consequence, the number of vertices represented in the logical diagram for a given Boolean Algebra is lower than that in its Hasse counterpart:

| diagram | # vertices || $\mathbb{B}_2$ | $\mathbb{B}_3$ | $\mathbb{B}_4$ | $\mathbb{B}_5$ |
|---------|-----------|----|----|----|----|
| *Hasse* | $2^n$ || $2^2 = 4$ | $2^3 = 8$ | $2^4 = 16$ | $2^5 = 32$ |
| *logical* | $2^n - 2$ || $2^2 - 2 = 2$ | $2^3 - 2 = 6$ | $2^4 - 2 = 14$ | $2^5 - 2 = 30$ |

For $\mathbb{B}_3$, the contrast is between the 8 vertices of a 'cubic' Hasse diagram and the 6 vertices of a hexagonal logical diagram [1]. For $\mathbb{B}_4$, the contrast is between the 16 vertices of a 'hypercubic' Hasse diagram and the 14 vertices of the rhombic dodecahedron (RDH), a 3D logical diagram that has been studied intensively in Logical Geometry [17]. Recently, a 3D visualisation has been (re)discovered as the logical diagram for the 30 non-trivial vertices

of $\mathbb{B}_5$, in the form of the rhombic icosahedron (RIH) [25,26]. As argued in [27], 22 of these vertices are located on the convex hull of the RIH, and the remaining 8 are located on the inside of the RIH. In the following two subsections, various subdiagrams of this RIH for $\mathbb{B}_5$ will be considered in more detail.

*4.1. From $\mathbb{B}_5$ to $\mathbb{B}_4$ by Collapsing Bit Positions*

A first operation that allows us to establish a systematic connection between the bitstrings of length 5 in $\mathbb{B}_5$ and those of length 4 in $\mathbb{B}_4$ is based on redundancy: in those cases where two bit positions systematically have the same value, these two positions can be COLLAPSED into one. For bitstrings of length 5, there are 10 different ways ($C_2^5$) in which 2 out of the 5 bit positions $b_n$ for $1 \leq n \leq 5$ can have identical values, namely,

$$
\begin{array}{cccc}
b_1 = b_2 & b_1 = b_3 & b_1 = b_4 & b_1 = b_5 \\
— & b_2 = b_3 & b_2 = b_4 & b_2 = b_5 \\
— & — & b_3 = b_4 & b_3 = b_5 \\
— & — & — & b_4 = b_5
\end{array}
$$

In Table 1, the particular case is illustrated where $b_3 = b_5$. Since we only consider the 30 non-trivial bitstrings of $\mathbb{B}_5$, there are (only) 7 bitstrings in the second column for which $b_3 = b_5 = 1$—the trivial bitstring 11111 is missing from the bottom row—and similarly, (only) 7 bitstrings in the fifth column for which $b_3 = b_5 = 0$—the trivial bitstring 00000 is missing from the top row. The 16 bitstrings of length 5 that do not show up in Table 1 are of course those 16 for which $b_3 \neq b_5$.

**Table 1.** From $\mathbb{B}_5$ to $\mathbb{B}_4$ by collapsing bit positions 3 and 5.

| $\mathbb{B}_4$ Collapse to $b_3$ | $\mathbb{B}_5$ $b_3 = b_5 = 1$ | $\mathbb{B}_4$ Collapse to $b_5$ | $\mathbb{B}_4$ Collapse to $b_3$ | $\mathbb{B}_5$ $b_3 = b_5 = 0$ | $\mathbb{B}_4$ Collapse to $b_5$ |
|---|---|---|---|---|---|
| 0010 | 00**1**0**1** | 0001 | — | — | — |
| 0011 | 00**1**1**1** | 0011 | 0001 | 00**0**1**0** | 0010 |
| 0110 | 01**1**0**1** | 0101 | 0100 | 01**0**0**0** | 0100 |
| 0111 | 01**1**1**1** | 0111 | 0101 | 01**0**1**0** | 0110 |
| 1010 | 10**1**0**1** | 1001 | 1000 | 10**0**0**0** | 1000 |
| 1011 | 10**1**1**1** | 1011 | 1001 | 10**0**1**0** | 1010 |
| 1110 | 11**1**0**1** | 1101 | 1100 | 11**0**0**0** | 1100 |
| — | — | — | 1101 | 11**0**1**0** | 1110 |

Now the collapse from $\mathbb{B}_5$ to $\mathbb{B}_4$ can take place in two directions: in columns one and four of Table 1, the two identical values collapse to the left position of $b_3$ while $b_5$ 'disappears', whereas in columns three and six, they collapse to the right position of $b_5$ while $b_3$ 'disappears'. Both directions of collapsing—columns 1/4 versus columns 3/6—yield the same set of 14 non-trivial bitstrings of length 4, corresponding to the 14 vertices of the RDH. In other words, this type of collapsing operation does not lead to the reappearance of the trivial top and bottom elements 1111 and 0000 of $\mathbb{B}_4$.

Visually speaking, this operation yields a collapse from the 30 vertices of the rhombic icosahedron (RIH) to the 14 vertices of an embedded rhombic dodecahedron (RDH). Figure 2—which has been built by means of the open source virtual reality modelling language X3D (Extensible 3D) and rendered by means of the Octaga Player tool—illustrates the particular case described in Table 1, i.e., for $b_3 = b_5$. The seven vertices on the (yellow) top left 'shell' of the RIH in Figure 2 are precisely decorated by the seven bitstrings from the second column of Table 1 for which $b_3 = b_5 = 1$. By way of mirror image, the seven vertices on the (yellow) bottom right 'shell' of the RIH are decorated by the seven bitstrings from the fifth column of Table 1 for which $b_3 = b_5 = 0$. In other words, this collapsing operation eliminates the 16 vertices which are located on the 2 'cubic' structures in the middle of the RIH, namely, the 8 vertices on the left hand (green) cube—for which $b_3 = 1 \neq 0 = b_5$—and

the 8 vertices on the right hand (red) cube—for which $b_3 = 0 \neq 1 = b_5$. Fitting together the $2 \times 7$ vertices of the 2 yellow shells in Figure 2 yields the 14 vertices of an embedded RDH.

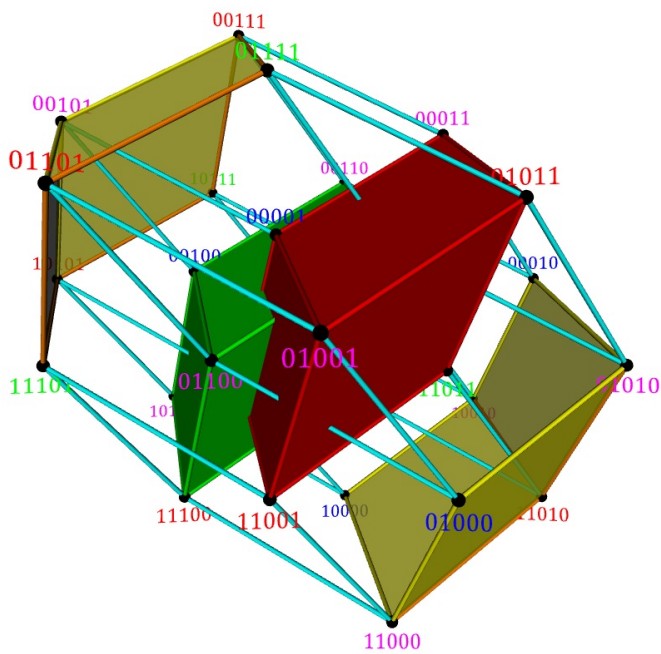

**Figure 2.** RDH inside RIH by collapsing bit position 3 and 5.

*4.2. From $\mathbb{B}_5$ to $\mathbb{B}_4$ by Deleting Bit Positions*

The second operation that allows us to establish a systematic connection between the bitstrings of length 5 in $\mathbb{B}_5$ and those of length 4 in $\mathbb{B}_4$ is simpler and more direct than the first one. It does not concern constraints on values for different bit positions, but instead simply DELETES a bit position. Obviously, for bitstrings of length five, there are exactly five ways to delete a bit position $b_n$ for $1 \leq n \leq 5$ in order to go from $\mathbb{B}_5$ to $\mathbb{B}_4$:

|  | delete $b_1$ | delete $b_2$ | delete $b_3$ | delete $b_4$ | delete $b_5$ |
|---|---|---|---|---|---|
| $\mathbb{B}_5$ | **1**0101 | 1**0**101 | 10**1**01 | 101**0**1 | 1010**1** |
| $\mathbb{B}_4$ | 0101 | 1101 | 1001 | 1011 | 1010 |

In Table 2, the particular case is illustrated where $b_1$ is deleted. Again, we only consider the 30 non-trivial bitstrings of $\mathbb{B}_5$: the trivial bitstring 00000 is missing from the third column in the top row and the trivial bitstring 11111 is missing from the fourth column in the bottom row. The general idea is that, on any given row, the $\mathbb{B}_5$ bitstrings in columns one and three, as well as the ones in columns four and six, only differ from one another with respect to their first bit position—in particular $b_1 = 1$ and $b_1 = 0$, respectively. By deleting precisely this first bit position, these $\mathbb{B}_5$ bitstrings thus pairwise collapse into the single $\mathbb{B}_4$ bitstring that is located in between them in columns two and five. In contrast to the situation depicted for the collapsing operation in Table 1, however, this deletion operation in Table 2 *does* result in the reappearance of the trivial top and bottom elements 1111 and 0000 of $\mathbb{B}_4$.

Visually speaking, this deletion operation yields a collapse from the 30 vertices of the rhombic icosahedron (RIH) to the 16 vertices of an embedded hypercube. Figure 3 illustrates the particular case described in Table 2, i.e., for the deletion of $b_1$. The eight vertices on the (green) right hand 'cubic' substructure of the RIH in Figure 3 are precisely decorated by the eight bitstrings from the first column of Table 2, for which $b_1 = 1$. By way of mirror image, the eight vertices on the (red) left hand 'cubic' substructure are decorated by the eight bitstrings from the sixth column of Table 2, for which $b_1 = 0$. In other words, this deletion operation eliminates the 14 vertices which are located on the 'flattened' RDH structure in the middle of the RIH. Fitting together the $2 \times 8$ vertices of these red and green

cubic substructures—as is indicated by means of the extra thick (magenta) edges through the center in Figure 3—yields the 16 vertices of an embedded hypercube.

**Table 2.** From $\mathbb{B}_5$ to $\mathbb{B}_4$ by deleting bit position 1.

| $\mathbb{B}_5$ $b_1 = 1$ | $\mathbb{B}_4$ Delete $b_1$ | $\mathbb{B}_5$ $b_1 = 0$ | $\mathbb{B}_5$ $b_1 = 1$ | $\mathbb{B}_4$ Delete $b_1$ | $\mathbb{B}_5$ $b_1 = 0$ |
|---|---|---|---|---|---|
| **1**0000 | 0000 | — | **1**0001 | 0001 | **0**0001 |
| **1**0010 | 0010 | **0**0010 | **1**0011 | 0011 | **0**0011 |
| **1**0100 | 0100 | **0**0100 | **1**0101 | 0101 | **0**0101 |
| **1**0110 | 0110 | **0**0110 | **1**0111 | 0111 | **0**0111 |
| **1**1000 | 1000 | **0**1000 | **1**1001 | 1001 | **0**1001 |
| **1**1010 | 1010 | **0**1010 | **1**1011 | 1011 | **0**1011 |
| **1**1100 | 1100 | **0**1100 | **1**1101 | 1101 | **0**1101 |
| **1**1110 | 1110 | **0**1110 | — | 1111 | **0**1111 |

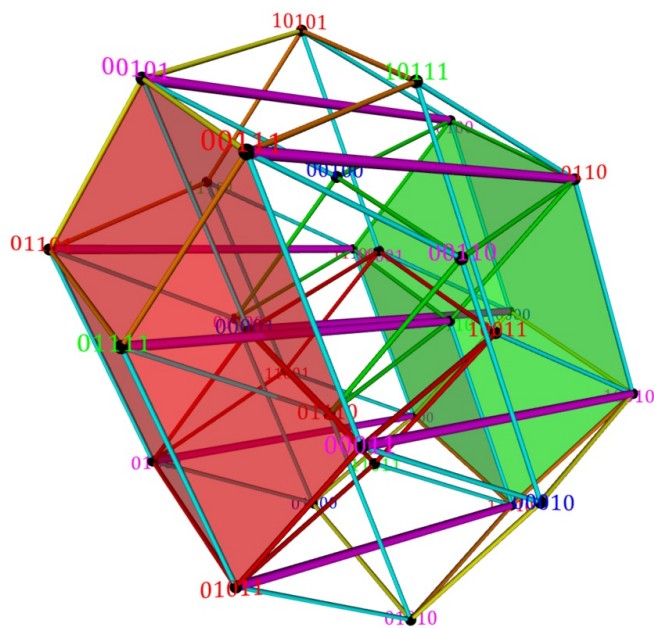

**Figure 3.** Hypercube inside RIH by deleting bit position 1.

## 5. Aristotelian Subdiagrams For B4

In this section, we briefly want to demonstrate how the distinction between the two transformations from $\mathbb{B}_5$ to $\mathbb{B}_4$—namely, that of COLLAPSING bit positions (Section 4.1) and that of DELETING bit positions (Section 4.2)—sheds an interesting new light upon the relationship between $\mathbb{B}_4$ and $\mathbb{B}_3$ as well.

### 5.1. From $\mathbb{B}_4$ to $\mathbb{B}_3$ by Collapsing Bit Positions

Remember from Section 4.1 that—when two bit positions systematically have the same value—a situation of redundancy arises, and these two positions can be COLLAPSED into one. For bitstrings of length four, there are six different ways ($C_2^4$) in which two out of the four bit positions $b_n$ for $1 \leq n \leq 4$ can have identical values, namely:

$$
\begin{array}{ccc}
b_1 = b_2 & b_1 = b_3 & b_1 = b_4 \\
— & b_2 = b_3 & b_2 = b_4 \\
— & — & b_3 = b_4
\end{array}
$$

**Table 3.** From $\mathbb{B}_4$ to $\mathbb{B}_3$ by collapsing bit positions 1 and 4.

| $\mathbb{B}_3$ Collapse to $b_1$ | $\mathbb{B}_4$ $b_1 = b_4 = 1$ | $\mathbb{B}_3$ Collapse to $b_4$ | $\mathbb{B}_3$ Collapse to $b_1$ | $\mathbb{B}_4$ $b_1 = b_4 = 0$ | $\mathbb{B}_3$ Collapse to $b_4$ |
|---|---|---|---|---|---|
| 100 | **1**00**1** | 001 | — | — | — |
| 101 | **1**01**1** | 011 | 001 | **0**01**0** | 010 |
| 110 | **1**10**1** | 101 | 010 | **0**10**0** | 100 |
| — | — | — | 011 | **0**11**0** | 110 |

In Table 3, the particular case is illustrated where $b_1 = b_4$. Since we only consider the 14 non-trivial bitstrings of $\mathbb{B}_4$, there are (only) 3 bitstrings in the second column for which $b_1 = b_4 = 1$—the trivial bitstring 1111 is missing from the bottom row—and similarly, (only) 3 bitstrings in the fifth column for which $b_1 = b_4 = 0$—the trivial bitstring 0000 is missing from the top row. The 8 bitstrings of length 4 that do not show up in Table 3 are of course those 8 for which $b_1 \neq b_4$.

Now the collapsing from $\mathbb{B}_4$ to $\mathbb{B}_3$ can take place in two directions: in columns one and four of Table 3, the two identical values collapse to the left position of $b_1$ while $b_4$ 'disappears', whereas in columns three and six, they collapse to the right position of $b_4$ while $b_1$ 'disappears'. Both directions of collapsing—columns one/four versus columns three/six—yield the same set of six non-trivial bitstrings of length three, corresponding to the six vertices of a (strong) Jacoby–Sesmat–Blanché hexagon. In other words, this type of collapsing operation does not lead to the reappearance of the trivial top and bottom elements 111 and 000 of $\mathbb{B}_3$.

Visually speaking, the six different ways to collapse two bit positions in $\mathbb{B}_4$ correspond to the well-known embedding of six Jacoby–Sesmat–Blanché (JSB) hexagons inside the RDH [1]. Furthermore, the complements of these six hexagons are polyhedra with eight vertices, which are called RHOMBICUBES and which—from an Aristotelian perspective—are Buridan octagons [11]. The diagram in the middle of Figure 4 illustrates this relationship of complementarity between the JSB hexagon on the left and the Buridan rhombicube on the right.

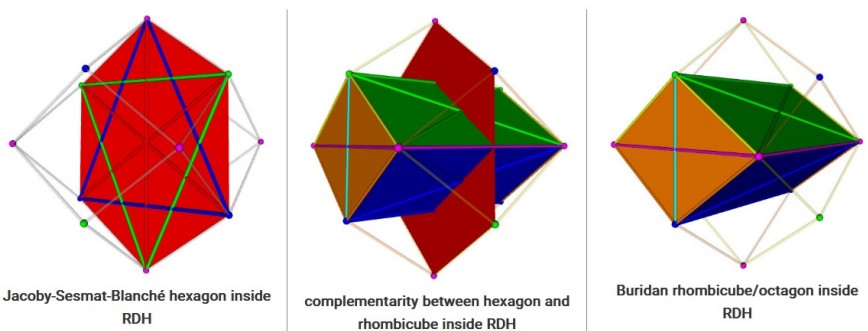

Jacoby-Sesmat-Blanché hexagon inside RDH     complementarity between hexagon and rhombicube inside RDH     Buridan rhombicube/octagon inside RDH

**Figure 4.** JSB hexagon and rhombicube inside RDH.

*5.2. From $\mathbb{B}_4$ to $\mathbb{B}_3$ by Deleting Bit Positions*

As was the case in Section 4.2, the second operation does not concern constraints on values for different bit positions, but instead simply DELETES a bit position. Obviously, for bitstrings of length four, there are exactly four ways to delete a bit position $b_n$ for $1 \leq n \leq 4$ in order to go from $\mathbb{B}_4$ to $\mathbb{B}_3$:

|  | delete $b_1$ | delete $b_2$ | delete $b_3$ | delete $b_4$ |
|---|---|---|---|---|
| $\mathbb{B}_4$ | **1**010 | 1**0**10 | 10**1**0 | 101**0** |
| $\mathbb{B}_3$ | 010 | 110 | 100 | 101 |

In Table 4, the particular case is illustrated where $b_1$ is deleted. Again, we only consider the 14 non-trivial bitstrings of $\mathbb{B}_4$: the trivial bitstring 0000 is missing from the third column in the top row and the trivial bitstring 1111 is missing from the fourth column in the bottom

row. On any given row, the $\mathbb{B}_4$ bitstrings in columns one and three, as well as the ones in columns four and six, only differ from one another with respect to their first bit position—in particular $b_1 = 1$ and $b_1 = 0$, respectively. By deleting precisely this first bit position, these $\mathbb{B}_4$ bitstrings thus pairwise collapse into the single $\mathbb{B}_3$ bitstring that is located in between them in columns two and five. In contrast to the situation depicted for the collapsing operation in Table 3, however, this deletion operation in Table 4 *does* result in the reappearance of the trivial top and bottom elements 111 and 000 of $\mathbb{B}_3$.

**Table 4.** From $\mathbb{B}_4$ to $\mathbb{B}_3$ by deleting bit position 1.

| $\mathbb{B}_4$ $b_1 = 1$ | $\mathbb{B}_3$ Delete $b_1$ | $\mathbb{B}_4$ $b_1 = 0$ | $\mathbb{B}_4$ $b_1 = 1$ | $\mathbb{B}_3$ Delete $b_1$ | $\mathbb{B}_4$ $b_1 = 0$ |
|---|---|---|---|---|---|
| **1**000 | 000 | — | **1**100 | 100 | **0**100 |
| **1**001 | 001 | **0**001 | **1**101 | 101 | **0**101 |
| **1**010 | 010 | **0**010 | **1**110 | 110 | **0**110 |
| **1**011 | 011 | **0**011 | — | 111 | **0**111 |

Visually speaking, this type of deletion operation yields the hithertho unnoticed embedding of a new type of polyhedron inside an RDH, as illustrated in Figure 5. We will refer to this subdiagram of an RDH as a RHOMBIC HEXAHEDRON, since it has eight vertices and six rhombic faces. It looks like a 'squeezed cube', and in this respect closely resembles the (more familiar) rhombicube at the right in Figure 4.

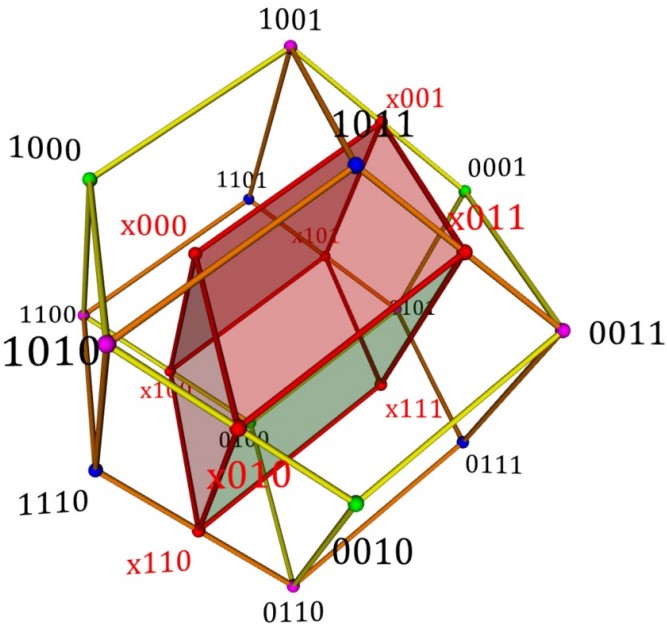

**Figure 5.** Rhombic hexahedron inside the rhombic dodecahedron.

## 6. Conclusions

The Boolean algebra $\mathbb{B}_5$—with its bitstrings of length five—has played the central role in this paper. On a more descriptive level, $\mathbb{B}_5$ was shown to underlie the semantic analysis of both proportional quantification with fractions and percentages—as in *two thirds/66 percent of the children are asleep*—and normative quantification—as in *not enough/too many children are asleep*. Furthermore, with both types of quantification, the pentapartite scalar structure could be modified by having bit positions collapse either at the extreme ends of the scale or in the very center of the scale, triggering reductions to $\mathbb{B}_4$ and $\mathbb{B}_3$, respectively.

On a more theoretical level, we have studied two distinct Aristotelian subdiagrams in $\mathbb{B}_5$. The operation of collapsing 2 bit positions with identical values allowed us to shift from the 30 vertices of the rhombic icosahedron RIH for $\mathbb{B}_5$ to the 14 vertices of an

embedded rhombic dodecahedron RDH for $\mathbb{B}_4$. The operation of deleting a bit position, by contrast, allowed us to shift from the 30 $\mathbb{B}_5$ vertices of the RIH to the 16 $\mathbb{B}_4$ vertices of an embedded hypercube.

With the corresponding collapsing operation for shifting from $\mathbb{B}_4$ to $\mathbb{B}_3$, the 14 $\mathbb{B}_4$ vertices of an RDH collapsed into the 6 $\mathbb{B}_3$ vertices of a strong Jacoby–Sesmat–Blanché hexagon. This is a well-studied phenomenon in Logical Geometry, including its complementarity with the Buridan octagons of the rhombicube. Going back up to $\mathbb{B}_5$ again, however, one question for further research concerns the precise Aristotelian properties of the so-called hypercube that serves as the complement of an RDH inside an RIH at the bottom right corner in Table 5.

**Table 5.** Complementarities in $\mathbb{B}_4$ and $\mathbb{B}_5$.

| 6 JSB hexagons in RDH with 6 vertices | complementarity in $\mathbb{B}_4$ | 6 rhombicubes in RDH with 8 vertices |
| --- | --- | --- |
| 10 RDHs in RIH with 14 vertices | complementarity in $\mathbb{B}_5$ | 10 'hypercubes' in RIH with 16 vertices |

With the deletion operation for shifting from $\mathbb{B}_4$ to $\mathbb{B}_3$, the 14 $\mathbb{B}_4$ vertices of an RDH collapsed into the 8 $\mathbb{B}_3$ vertices of a hithertho unnoticed type of polyhedron inside an RDH, which we have dubbed a RHOMBIC HEXAHEDRON, and which is illustrated in Figure 5. The operation of deleting a bit position (both from $\mathbb{B}_4$ and $\mathbb{B}_5$) can be straightforwardly related to constellations in which a given bit position—in addition to its standard values 0 and 1—may also be left un(der)specified. In Logical Geometry, the notion of a PROTO-BITSTRING has on occasion been put forward for this phenomenon. In the light of the discussion in the present paper, we strongly plead for this topic of proto-bitstrings to be investigated more thoroughly in future work.

**Author Contributions:** Conceptualization, K.R. and H.S.; Formal analysis, K.R. and H.S.; Investigation, K.R. and H.S.; Writing—original draft, H.S.; Visualization, H.S. All authors have read and agreed to the published version of the manuscript.

**Funding:** This research was funded by the research project 'BITSHARE: Bitstring Semantics for Human and Artificial Reasoning' (IDN-19-009, Internal Funds KU Leuven).

**Data Availability Statement:** Not applicable.

**Acknowledgments:** A previous version of this paper was presented at the Seventh World Conference on the Square of Opposition (KU Leuven, 2022). In addition to the audience of this talk, the authors would like to thank Lorenz Demey for his detailed comments and suggestions, as well as the three anonymous reviewers for their valuable feedback.

**Conflicts of Interest:** The authors declare no conflict of interest. The funders had no role in the design of the study; in the collection, analyses, or interpretation of data; in the writing of the manuscript, or in the decision to publish the results.

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
