# Peer review of "Aristotelian Fragments and Subdiagrams for the Boolean Algebra 5"

_axioms, doi:10.3390/axioms12060604_

Round 1
Reviewer 1 Report
In my opinion, the results given in this article may provide a theoretical model for natural language processing and may have some impact on logical geometry. The paper is written well and presentation is very clear. The figures in the paper are very beautiful. But I think the author should indicate which program environment was used for drawing these figures.
Author Response
We thank the reviewer for their appreciative words. We have added a reference to the open source virtual reality modelling language X3D (Extensible 3D) format and the Octaga Player rendering tool on p.10 where the first 3D figure, nl. Figure 2 is discussed.
Reviewer 2 Report
The paper is a relevant and very well written contribution to logical geometry and theory of quantification, in accordance with all scholarly standards, with clear presentation and precise argumentation. The proportional and normative quantification (e.g., fractions and percentages, and "not enough", "too many", respectively) is analysed by means of Boolean algebra B_5 (bitstrings length 5). The consequences of the collapse and deletion of bit positions are explored and presented using Aristotelian subdiagrams of B_5. A new type of logical polyhedron is discovered.
I have only several typographical/lexical remarks and a comment:
Page 2 line 51: "logically possible combinations" may stay informally; of course, variations with repetition are meant.
Page 2, line 59 (in \mathdisplay): the use of \langle and \rangle can be recommended instead of < and >.
Page 2 line 55, page 3 line 79, page 4, line 133: it could be better (non-confusing) to consistently use non-bold for bit values (it seems that sometimes bold is used, e.g., in the last row, for bit values).
Page 3, line 104: maybe to put "generalized quantification theory" or similar in parentheses after the first occurrence of 'GQT'.
Page 6, lines 185-186: the right curly bracket is missing.
Page 7, lines 199-200: the right curly bracket is missing.
Page 7, line 216: the authors' remark "check" remained unsolved.
A comment on page 9, lines 301-310: However, it seems it could also be said that (inverted) Hasse diagram omits the top and bottom elements of Seismat-Blanché hexagon (p&q v -p&-q, (pvq)&(-pv-q) (their position should be in the middle area of a Hasse diagram).
